# The Tarteel Dataset: Crowd-Sourced and Labeled Quranic Recitation

**Hamzah I. Khan**[*]
Tarteel AI
hamzah@tarteel.ai

**Anas Abou Allaban**[*]
Tarteel AI
anas@tarteel.ai

**Mohamed Medhat Moussa**[*]
Tarteel AI
mohamed@tarteel.ai

**Abubakar Abid**
Stanford University
a12d@stanford.edu

## Abstract

We propose a standard schema for paired Quranic audio and text datasets. We describe the collection, labeling, and validation of the Tarteel recitation dataset, the first large-scale dataset of Quranic recitation and accompanying Arabic text collected in a crowd-sourced manner. The dataset contains 25,000 audio clips totalling 67.39 hours of audio and represents a wide variety of recitation styles, proficiencies, and speeds. The data were collected over a period of six months from over 1,200 unique individuals of different ages, genders, and ethnicities. We describe the composition of the data and contributors, describe in detail how the data was collected and processed, and give some baseline performance for preliminary machine learning algorithms that were trained and evaluated on the dataset.

## 1 Introduction

The Quran is held by Muslims to be a sacred book, and its recitation in Arabic is an important practice in the faith of 1.8 billion Muslims around the world [18]. The Quran is recited as part of daily prayers in Islam, and its recitation and memorization is encouraged from an early age through classes and competitions. Traditionally, the recitation of the Quran has been taught to students in-person by teachers and *imams* [1]. There are a number of ways in which the recitation of the Quran, though in Arabic, differs from ordinary spoken Arabic. Some of these stylistic differences are codified in rules known as *tajweed* [2].

Given the important place of reciting the Quran in the lives of Muslims, it is of increasing interest to build automated software tools that can help Muslims recite the Quran with greater precision [15, 1]. In order to build many of these applications, accurate deep learning models that recognize and understand the recitation of the Quran are needed. And in order to train such models, a high quality dataset covering a diverse range of voices and recitation styles of the Quran is needed – that is the focus and contribution of this paper.

Our goal here is to collect a *crowd-sourced*, large-scale dataset of Quranic recitations that are annotated with the Arabic text of the verse that is being recited. Each audio file in the dataset consists of the recitation of a single verse. In this regard, it parallels datasets of professional Quranic recitations that are widely available online. However, as we discuss in Section 2, there are key differences between the Quranic recitation of ordinary Muslims and audio from professional Quranic reciters, which motivate the collection of the Tarteel recitation dataset.

---

[*]equal contributor

35th Conference on Neural Information Processing Systems (NeurIPS 2021), Sydney, Australia.

The Tarteel dataset is very much a collected "in-the-wild" dataset; it includes real-world noise and artifacts, which were not removed or preprocessed. This makes the dataset particularly useful for developing models and applications that are meant to be deployed to users in the real world. At the same time, we have carried out certain quality assessment steps to verify that the labels for the collected verses are correct. These are described in Section 3 and 4.

Our hope is that the Tarteel[2] recitation dataset spurs many new machine learning algorithms for the recognition and correction of Quranic recitation, which can now be measured against real audio from ordinary reciters of the Quran. We have developed and tested preliminary machine learning models on the dataset; the results are described in Section 6, and we lay out steps for future work in Section 7.

The URLs to the audio files and annotations can be downloaded from the Tarteel website: `https://www.tarteel.ai/dataset`.

## 2 Background

### 2.1 Related Works

A variety of Arabic speech datasets have been collected and published in recent years, for various tasks including but not limited to Automatic Speech Recognition, Speech Synthesis, Emotion detection and Dialect Identification. The GALE speech dataset [10] released by the Linguistic Data Consortium provides 37 hours of Arabic speech extracted from broadcast news channels. The dataset consists of both male and female speakers, and is relatively clean acoustically as the audio was recorded in a studio setting. The MGB-2 dataset [3] is another dataset collected in a similar setting (broadcast news) but is considerably larger with 1200 hours of speech. The Arabic Speech Corpus [13] is another dataset recorded in a professional studio, primarily for the purposes of building a speech synthesis system. Conversely, several speech datasets have also been collected in a non-professional and noisy environment, as this kind of setting is much more relevant for many user-facing speech models and applications. The King Saud University Arabic Speech Database [5] is one such dataset with 590 hours of speech from speakers with diverse backgrounds, ethnicities and genders. Two-thirds of the dataset has been collected from real-world non-studio settings. The MGB-3 [4] and Arabic Natural Audio [16] datasets also provide speech corpora in a similar vein, with speech of varying quality extracted from YouTube channels. Finally, the Mozilla Common Voice Project [3] is another project that aims to collect audio from noisy real-world settings. Any user can visit the website and contribute their voice regardless of the environment and equipment they are using. This project is not currently available in Arabic, but has a very similar approach to corpus building as our work.

General speech corpora like the ones described above still present a significant hurdle for the problem targeted by this work; spoken Arabic, as one might use in every-day life, is very different in style and vocabulary than Quranic recitation. A number of datasets are available online of Quranic recitation by professional reciters [7], however these reciters have perfected their recitation quality and style, and are seldom comparable to ordinary recitation. The vast majority of ordinary recitation has properties that are not exhibited in these datasets, like hesitations, dialectal differences, phonetic co-articulation artifacts and differences in elongation styles. [14] provides a comprehensive overview of these issues. Finally, some other datasets have been released for other languages exploring similar non-general speech like acapella singing [21, 8, 11].

### 2.2 The EveryAyah Dataset

One existing dataset of interest in this area is the EveryAyah Dataset [7], which contains recitation audio from 26 professional reciters. The dataset contains, for every verse in the Quran, one recording from each reciter, as well as an image of the Arabic script of the verse in both low resolution and high resolution format. Some of the recitation audio is labeled with timing data that indicated the timestamp within the audio file at which each word is recited. The data is served from a web page from which a user can query a reciter, a chapter number, and a verse number from that chapter and receive the audio recording, the images of the script of the verse, and timing information if it exists.

---

[2]The name "tarteel" comes from the Quran itself, where Muslims are instructed to "recite the Quran with *tarteel* (slow, measured rhythmic tones)" (Quran 73:4).

[3]`https://voice.mozilla.org`

Of note is that the EveryAyah Dataset does not provide a text label for the verse recited in each recording. Reciters sometimes repeat words while reciting. For example, a reciter may pause to take a breath before repeating the previous few words to provide continuity in the recitation. Because of this repetition, it can not necessarily be assumed that the correct label for a verse is the exact text of that verse. The EveryAyah Dataset is unlicensed.

## 2.3 Use Cases

We envision several potential applications that can make use of this recitation-style dataset. Automatic recitation recognition can transcribe the recitation from a person, providing a textual representation that can be used downstream for purposes of tracking progress through the Quran, subtitling for increasing accessibility for people with hearing impairments and helping people with memorization by checking if the text matches the verses in the Quran. The dataset can also be used to train Computer Aided Language Learning systems that check pronunciation and the aforementioned *tajweed* rules. Other applications from general speech systems like speaker identification, gender detection, etc. can also be improved in the recitation domain by using this domain-specific dataset. Finally, the dataset can also aid in improving existing non-general speech systems (recitation-, poetry-, singing-based models) by exploring architectures that are more suited to the subtleties present in these kinds of audio.

## 3 A Proposal for a standardized Quranic dataset schema

Given the unique nature of Quranic speech recognition, we propose a new dataset schema that can be used as a standard for future Quranic audio datasets. Many existing Quranic audio datasets are simply folders of audio data, which may require significant pre-processing for use in data-driven development. A common standard such as the one we propose applies our learnings during our data collection process and should improve dataset usability.

Each recording in a Quranic audio dataset should be associated with a row containing all of the fields in Figure 1. Each row would contain the URL of the associated recording file as well as a unique ID. While not necessary, the suggested naming convention of the file and unique ID would be: `<ChapterNumber>_<VerseNumber>_<UUIDv4>`, where the file name is the same as the unique ID except it includes the file extension (ex. `.wav`, `.mp3`, etc.).

Each row would also contain two types of labels: an Arabic text label of the speech in the recording as well as a phoneme label. As *tajweed* pertains to the pronunciation and elongation of the recitation, text labels are not sufficient to identify whether a verse of the Quran was recited with correct *tajweed*. We do not propose a particular type of *tajweed* label in this work, but include it in our proposed schema.

Each row also contains four pieces of optional metadata: age, heritage, gender, and recitation method (*qira'ah*). If the information is unavailable, then the field should be filled with a null value. The reasons for the inclusion of this information for each recording are further described in Section 3.1.

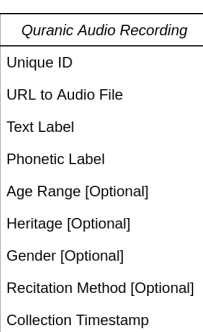

Figure 1: A proposed schema for rows in Quranic audio recitation datasets.

### 3.1 Diversity & Representation

Collecting data meant to be used for technology that works for everyone requires collecting a varied dataset representative of multiple demographics within the Muslim population. At the same time, we must balance this data collection with the need for user privacy. Hence, the proposed schema *optionally* includes user demographic information.

Furthermore, we suggest that dataset designers consider asking users to provide an age range as opposed to a specific age. While this may reduce accuracy in demographic collection, we believe that the privacy of users should come first, especially when it comes to religious affairs pertaining, for example, to recitations by women. Respecting these religious practices requires adhering to certain privacy measures.

We also want to clarify our choice of including heritage in the schema as opposed to nationality. In the context of Quranic recitation, a person's background is generally more important than their actual nationality since it identifies the Arabic dialect they are more accustomed to speaking. Different dialects have different vocal nuances and it is important to identify the vocal trends associated with a certain dialect or region.

With regard to the recitation method, there are 7 oral chains of narration for the Quran with the chain of *Hafs* being the most common around the world [19].

## 4 How the Tarteel Dataset Audio was Collected

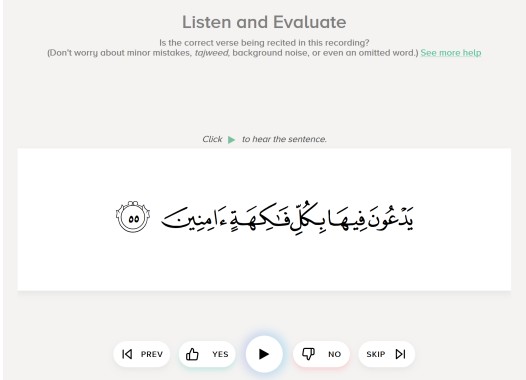

Figure 2: An example of the evaluator available on Tarteel.io. Users can evaluate the recitation as correct or incorrect by clicking Yes or No and have the option to skip evaluations if they are unsure.

In this section, we describe how we collected and labeled the Tarteel dataset.

### 4.1 Tarteel Dataset Schema

The Tarteel dataset consists of 25,000 audio clips containing over 67 hours of recitation audio and representing a wide variety of recitation styles, proficiencies, and speeds. The data were collected over a period of six months from over 1,200 unique individuals of different ages, genders, and ethnicities. Each recording is associated with a single row containing 11 pieces of information about the recording in a CSV file available for download at https://www.tarteel.ai/dataset.

As described in the schema, each row contains the *surah* (chapter) number and the verse number of the recited verse, as well as a unique id (present in the audio filename) and URL from which the raw audio can be downloaded. Each row also contains a timestamp of when the recording was created as well as an indication of what mode of recitation on the Tarteel website (i.e. one at a time or continuous recitation) was used to create the recording.

Users are encouraged to complete an optional demographic profile so that the demographic fields in the dataset can be populated. If users choose not to submit demographic information, then the field is filled with a negative default value. We currently ask users to recite only from the narration chain

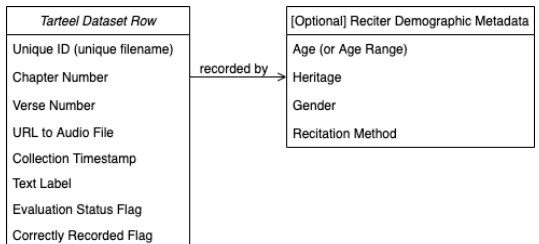

Figure 3: The schema for rows in the Tarteel Quranic Audio Recitation Dataset.

of *Hafs*, which is the most common chain. The privacy policy for collection is further discussed in Section 4.4.

Each row in the dataset also includes two fields that are not present in the proposed schema in Section 3 but are relevant for internal Tarteel operations: whether the recording has been evaluated and whether the verse was correctly recorded.

## 4.2 Obtaining audio recitation

The Tarteel dataset was designed from the beginning to be representative of the larger Muslim population, whose recitation may be incorrect or somewhat weaker than professional reciters. To this end, we developed a website to crowd source data collection, where users can either recite a verse that is displayed to them or select a specific verse they want to recite. By default, we display verses with the least amount of recordings sorted in the order of their appearance in the Quran. Transliteration is also available to support non-native Arabic readers. Participants were not paid to submit recitations.

## 4.3 Diversity analysis

One of the issues with our dataset is the limited diversity of recitations. There are over 10 different dialects of Arabic and 7 different recitation styles of the Quran. The majority of our data however comes from Egyptian users and all of it is recited in the style of *Hafs*. We hope to expand upon our dataset in future iterations with a global sample of users and include at least the next major recitation style, *Warsh*.

## 4.4 Privacy Policy and Ethical Considerations

As part of Tarteel's commitment to user privacy, we do not collect any personal identifiable information from our users. We only collect a user's submitted audio recitation and their demographic information, if they provided it. We clearly state in our policy and in supplementary material that this data will be distributed publicly under the CC BY-NC-ND 4.0 license.

Given the religious nature of the project, we found that Muslim users are naturally inclined to support our data collection efforts to the best of their abilities. This is because our data collection efforts promote developing technology that can potentially improve Muslims' relationship with their holy book and faith, a goal that resonates with many practicing Muslims.

## 5 How the Tarteel Dataset was Labeled

We tried two forms of labeling: crowd-sourced label verification and 'automated' label generation.

## 5.1 Crowd-Sourced Label Verification of the Tarteel Dataset

Since each of the recordings obtained is supposed to correspond to a given chapter and verse number, the label for that recording ideally should be the text of the verse identified by those numbers. This means that all of the recordings automatically can be assigned a label. However, given the crowd-

sourced nature of the dataset and the varying levels of contributor proficiency, it is necessary to guard against both malicious contributions as well as well-intentioned contributions riddled with mistakes.

### 5.1.1 Crowd-sourced binary evaluation

The proposed solution is a binary evaluation of whether the recording matches the expected verse being recited and whether it is complete. This high level evaluation is also crowd-sourced on our website where users are given the following rules for evaluation:

**1. Correct evaluations:**

- The reciter makes a mistake in how they pronounce a letter, or if he/she makes mistakes in harakaat (vowels).
- The reciter makes tajweed-related mistakes.
- The reciter makes a major mistake, but then corrects himself or herself.
- The recitation includes noise or an out-of-place word before or after the verse.
- The reciter recites very slowly or very quickly.

**2. Incorrect evaluations:**

- The recording is empty or is only background noise.
- The wrong verse is recited.
- The verse is recited so softly that you cannot hear the recitation (make sure your device volume is set properly!)
- Multiple verses are recited in the same recording, even if they include the correct verse.
- The reciter omits more than a single word of a verse.

### 5.1.2 Limitations of crowd-sourced evaluation

This approach has several limitations. The most limiting is that it does not scale well, as it requires an amount of manual work exceeding the size of the dataset itself; verification in most cases involves listening to the entire recording at least once and then taking action, either approval or rejection, on the recording. Another limitation is that it does not account for malicious reviewers, who could mark correct evaluations as incorrect or vice versa. Similarly, it does not account for reviewers without Arabic proficiency who could fail to recognize mistakes in the recording or incorrectly label a correct recitation as having mistakes and being incorrect. The latter two limitations could be at least partially mitigated by having each recording reviewed by multiple reviewers and then using the most common evaluation, however, this would only contribute to the key limitation of scaling the evaluation approach.

### 5.2 Automated Label Generation of the Tarteel Dataset

Instead of attempting to verify that the dataset recordings match the verses they were supposed to be recorded for, a new approach for labeling is applied that ignores all existing context for the recording and derives the label purely from the audio data. This approach involves feeding the audio through an existing speech-to-text model to derive a preliminary label, fuzzy searching and aligning that preliminary label in the text of the Quran, and then further adjusting the matching Quran text to include the additional common phrases known as the *basmallah* (statement of truth) and *istiatha* (statement of refuge) [4] in the label if detected in the audio.

### 5.2.1 Preliminary label derivation through speech-to-text

In order to establish a preliminary label for each recording, a text transcript is inferred from its audio data using the Google Cloud Speech API. The API is configured to recognize the Arabic dialect of the United Arab Emirates ("ar-AE") and is also provided the most common phrases from the Quran as additional context. The transcript obtained from the API is deemed to be a preliminary label to build from.

---

[4]These are two phrases that are recited when beginning the recitation of the Quran or when moving from one chapter to another.

### 5.2.2 Force-alignment of speech-to-text transcription to the Quran

Each recording's preliminary label obtained from applying speech-to-text is then provided as the input query phrase to Tarteel's proprietary Quran fuzzy-search engine. The search engine identifies any matching segment of the Quran for the given query phrase, spanning from a partial verse up to multiple consecutive verses. More specifically, it is able to recognize full and partial verses, consecutive verses, and verses similar but not exactly matching the query phrase in the case of no exact matches. In the case of inexact similarity, the determination of similarity is dependent on a calculation involving the query phrase length, the matching verse's length, the Levenshtein distance between the two strings, and some constants. Ultimately, the search engine returns either a string of text coming directly from the Quran, or a null value, which is then adjusted using the context of the preliminary label to include the *basmallah* if present.

### 5.2.3 Limitations of automated label generation

In generating a label for each recording with the automated approach, each step introduces a significant limitation. When deriving a preliminary label through applying Google's speech-to-text model, there are a sizeable amount of errors present in the transcription due to the relative inaccuracy of the model in transcribing Quran recitations. In some instances of incorrect transcription where the recording itself does not contain mistakes, this error is mitigated by the force-alignment step, whereas in others it leads to the force-alignment step failing to match any Quran verse. In the force-alignment step, error is introduced due to the intentional support for fuzzy matching, which has the side effect of rejecting any mistakes that are made in the recording and correctly identified by the speech-to-text model.

### 5.2.4 Segmenting audio data

In our process of cleaning our audio data and labels, we attempted to exploit the fixed corpus of the Quran to remove samples that had a high probability of not coming from the Quranic text. Our hypothesis was that if the data is closer to the source of truth, then the label will be more accurate and the model will perform better. We segmented the labels into four sections:

- Exact: An exact or highly similar match of the label with a verse from the Quran.

- Single Match, Inexact Words: A match for the label was found based on a number of shared words between the matching verse and the label, but the number of mismatched words or the ordering of the shared words gives us lower confidence in our match.

- Multiple, Inexact Matches: Multiple matching verses in the Quran were found for the label, without any single verse having high enough similarity to be a definite match.

- No Match: No match with a verse from the Quran.

We used a custom search engine based on a set of organized lookup tables to perform queries against the Quran corpus and find relevant verses. Based on the results of the search query and the matching process, a segment from the list above was assigned to each label. Only the 'Exact' and 'Single Match, Inexact Words' (Single) segments were used to construct a filtered dataset.

We found that performing this segmentation simply decreased the total number of samples that we could use without actually improving the performance of the model. We believe this is because the model did not have enough samples to work with and did not provide enough diversity to generalize on new audio samples.

Table 1: Model Results

| Model | Val WER, % | Params, M | Steps, k |
|---|---|---|---|
| DS2-5x512 (Single) | 16.40 | 24.6 | 242.75 |

## 6 Experimental Design and Results

We trained two different model architectures generally used for speech recognition: DeepSpeech2 (DS2) [6] and QuartzNet (QN) [17] on the union of the Tarteel and EA datasets. DeepSpeech2's architecture is based on LSTMs while QuartzNet is a convolutional model. Both architectures use Connectionist Temporal Classification (CTC) Loss [12]. We experiment with different model depths, augmentations, and steps. Models were trained either on a single Nvidia RTX3090 GPU or 4 Nvidia A100 GPUs. The results are summarized in Table 2 [5]. For brevity, we use the following naming convention:

- DS2-<HiddenLayers>x<HiddenSize>-<Augmentation> for DeepSpeech2 models
- QN-<Blocks>x<Modules>-<Augmentation> for QuartzNet models

Table 2: Model Results

| Model | Val WER, % | Params, M | Steps, k |
|---|---|---|---|
| DS2-5x512 | 13.07 | 24.6 | 231.32 |
| DS2-5x512-SpecAug | 27.34 | 24.6 | 216.29 |
| DS2-5x1024 | 16.22 | 86.6 | 692.13 |
| QN-5x5 | 16.60 | 6.7 | 529.95 |
| QN-10x5 | 13.91 | 12.8 | 665.87 |
| QN-15x5 | 15.11 | 18.9 | 91.86 |
| QN-15x5-SpecCutAug | 9.87 | 18.9 | 172.88 |
| QN-5x5-SpecAug | 27.75 | 6.7 | 137.77 |

SpecCutAug refers to Spectral Cutout Augmentation [9], while SpecAug refers to frequency masked spectral augmentation [20].

We find that using a larger DS2 model results in worse WER compared to the smaller one. Our hypothesis is that the larger model does not have enough data to optimize the network. This is not the case with the QN model however, where a larger model results in better WER. Some changes to consider would be using different optimizers and learning rates to see if they improve the performance of the DS2 model.

An interesting observation is that spectral augmentation with frequency masking results in worse WER for both the DS2 and QN models, which is not what is reported commonly in the literature. We believe this could be due to the fact that we use the MP3 encoding instead of Wave for our audio files, a design decision influenced by our limited storage and compute capacities. Spectral cut augmentation however seems to improve the performance of WER and results in the best performing model overall, as reported by [17].

Future combinations to test include time domain augmentations like modifying the speed, pitch, and white noise in the audio.

## 7 Discussion & Future Work

We propose a standard dataset schema for Quranic speech recognition tasks that is suited for tackling problems in this space such as differences in recitation styles and dialects. We apply this schema to the construction of the Tarteel recitation dataset, the first large-scale crowd-sourced dataset of Quranic recitation audio and text labels. We describe our approaches and challenges in collecting, cleaning, and validating the data, highlighting what worked and what didn't work. Specifically, we found that segmenting the data based on a label's similarity with respect to the corresponding Quranic verses did not improve model performance. Augmenting the data however using spectral cut augmentation was found to significantly improve the performance of the model. Future modeling work includes investigating why larger LSTMs result and spectrogram augmentation with frequency masking result in a worse WER. The Tarteel dataset does not contain tajweed labels for proper identification of the

---

[5]We don't perform a comparison of all combinations due to limited compute resources at the time of writing this paper.

unique pronunciation rules of the Quran (*tajweed*). The best structure of these labels is still unclear and one possible approach we may explore is phoneme-based labeling.

### 7.1  Societal Implications

One poignant example of technology that could be developed using this dataset includes Quranic mistake correction for the masses. Teachers of the Quran spend much of their time on two tasks: teaching their students the proper recitation of the Quran and listening to students recite the Quran and correcting any mistakes in real-time. As the latter is a largely manual process that could stand to be automated by recent advances in NLP, technology developed on the Tarteel dataset or any other dataset using the proposed schema in Section 3 could free up Quran teachers to do more teaching. Simultaneously, listening sessions tend to be times for students of Quran to learn etiquette or religious lessons in an unstructured manner. As Quranic audio datasets and the technologies dependent on them are developed, teachers of Quran must be kept in mind as an impacted stakeholder.

## Acknowledgments

A large number of people contributed to the development of the original Tarteel platform, including Abdulrahman Al-Fozan, Abdellatif Abdelfattah, Ali Abid, and Ali Abdalla. We are thankful for their advice and contributions.

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
