# OpenReview forum: "The Tarteel Dataset: Crowd-Sourced and Labeled Quranic Recitation"
_NeurIPS.cc/2021/Track/Datasets_and_Benchmarks/Round1 — Submitted to NeurIPS 2021 Datasets and Benchmarks Track (Round 1)_

### Official Review · Reviewer_gmVt · 2021-07-02
**An interesting dataset, but more information is needed**

**Rating:** 6
**Confidence:** 4
**Clarity:** It's well written.

**Strengths:**

The dataset can be helpful for developing applications that can increase the accessibility of Quran for Muslims, such as subtitling and reception recognition.

**Weaknesses:**

The paper lacks information on the crowd-workers and the analysis of both crowd-sourced verification and automatic labelling. Without such information it is difficult to understand how good the resulting dataset is (quality) and how it is different from existing corpora. For instance, how many recordings are annotated as incomplete/incorrect/maliciously produced and how the authors deal with those recordings? Do the same reciters also perform evaluation tasks?






**Additional Feedback:**

In the paper the authors mention ordinary recitation is different from the professional one. However, I could not find the related analysis (or examples) on such matter, nor on the collected data. I believe that providing more details on this can facilitate future work on using the created dataset, for both Muslims and non-Muslims. For instance, some examples illustrating how the elongation styles vary.

**Correctness:**

Regarding crowd-workers, how are the proficiencies of the worker defined? In general, how much time did it take for individual worker to finish each recording (another way to understand if there is malicious contributions)?

**Documentation:**

It is not very clear how the data is collected. What are the instructions? Were the workers instructed to perform recitation in certain ways?

In section 4.1, "The Tarteel dataset...representing a wide variety of recitation styles, proficiencies and speeds." I would suggest elaborating more with examples on: what and how many styles, proficiencies and speeds are there in the dataset.

What is the demographic distribution of the crowd-workers. e.g., what are the age ranges/genders of the workers (surely from the contributors who provided their demographic)?

**Relation To Prior Work:**

The paper is well motivated and it discusses prior work in good detail.

**Summary And Contributions:**

This paper proposes a collection of Quranic audio recitation dataset annotated with Arabic text of verse information and the demographic data of reciters (if provided). The authors argue that the existing corpora by professional Quranic reciters cannot represent the characteristics of ordinary recitation by Muslims, such as elongation styles and vocabulary. Such differences motivate the creation of Tarteel recitation dataset, which can help develop recitation applications.

The dataset consists of 25K audio clips recorded by 1200+ volunteers covering various recitation styles and speeds. In terms of data quality, the recordings are verified through crowd-sourced binary evaluation (e.g., whether a recording is complete and matches the corresponding verse) and automatic labelling to filter out inappropriate ones. In the end, baselines on speech recognition using the created dataset are provided.

---

### Official Review · Reviewer_asXm · 2021-07-03
**a) The contribution is limited. b)  The impact of this dataset in downstream NLP applications is unclear. c) Why existing tools cannot be used to generate equivalent data is unclear.**

**Rating:** 4
**Confidence:** 5
**Clarity:** Yes.

**Strengths:**

- Dataset is much larger than the existing data
- A lot of variety due to a large number of reciters.

**Weaknesses:**

a) The contribution is limited to only one particular book. Ideally one would like to have such an exercise on a large number of Arabic texts to establish impact and generalizability.
b) More details on the gathering policy are required. For instance, how many verses from a single user were recorded in a day? What was the gap between each recording? What was the average length of a verse? Were there exceptionally long verses? Was care taken to not overstress the reciter?
b)  The impact of this dataset in downstream NLP applications is unclear. For instance, does this data improve ASR in general? How does the DS2 model trained on this data perform in the wild on any input? Further, why was the EA data added? How does DS2 perform in the wild for the author's data only (no EA data)?
c) Why a pipeline of existing NLP tools cannot be used to generate equivalent data (e.g., https://www.resemble.ai/) is unclear.

**Additional Feedback:**

Please see the weaknesses noted above and make an attempt address them.

**Correctness:**

-- If the submission is a dataset, it is constructed in a sound way?
More or less.
-- If it is a benchmark, are the evaluation methods and experiment design appropriate and performed correctly?
Very limited (see weaknesses)

**Documentation:**

A lot of information about the crowdsourced data gathering is missing.

**Ethics:**

No ethical concerns as such.

**Relation To Prior Work:**

Fairly well.

**Summary And Contributions:**

The authors in this paper create crowdsourced Quranic audio and text datasets. They also describe a schema, some cleaning steps, and baseline performance of speech recognizers.

There is no response from the authors. Hence I will keep the score the same.

---

### Official Review · Reviewer_JHKH · 2021-07-05
**Good step forward, but several details missing. Contribution not significant.**

**Rating:** 4
**Confidence:** 3

**Strengths:**

1. First large-scale dataset of this kind
2. The recitation audio is not limited to professional reciters
3. Good comparison with existing datasets
4. Diverse collection from over 1200 unique individuals of different ages, genders and ethnicities

**Weaknesses:**

1. Relevant to a very specific community
2. Since the participants were not paid, it makes the task of cleaning more challenging
3. The authors have mentioned a lot of details about what did not work or had limitations (for example crowd-sourced label verification, label segmentation into four categories, etc.). More details on what worked would have been appreciated
4. The major contribution seems to be only in collecting the data and making it available. This doesn't seem to be substantial enough for this venue.
5. The paper describes serious limitations of both crowd-sourced verification (5.1) and automated label generation (5.2). The reader is confused about what was finally used and how the quality was ensured in spite of serious limitations with both approaches.
6. Experiments doesn't seem to corroborate the quality or utility of the dataset
7. Unless I have missed, Table 1 doesn't seem to be referred from anywhere
8. Authors have talked about several interesting use cases of the dataset. It would have been great to see more experiments establishing the utility of the dataset in some of those use cases.
9. The two fields which are not present in the schema, but still included "for internal Tarteel operations" is not clear. How and where are those fields populated and used?
10. The text label and phoneme label requires more explanation.
11. The only experiment done seems to have combined Tarteel and EA datasets. Why so?
12. The demographic information is hardly present in the dataset available for download. Wonder what value it adds then?

**Additional Feedback:**

Provided in "weaknesses".

**Clarity:**

In general the paper is well written, but there are certain areas (as pointed out in the weaknesses) which could have been better explained, given that authors had more than half a page more available to them!

**Correctness:**

The claims made seem to be correct, but given the above weaknesses, I have a lot of concerns.

**Documentation:**

Detail on data collection and organization - acceptable
Availability and maintenance - acceptable. Maintenance is not clear in the paper, though.
A lot of experimental details like problem setup, hyper parameters, train-test splits etc. are missing


**Ethics:**

Not sure. The dataset pertains to a specific community of people.

**Relation To Prior Work:**

Yes. 2.1 and 2.2 seems OK.

**Summary And Contributions:**

1. The authors have created a dataset of quranic recitation audio paired with the text
2. They have proposed a schema for storing the data
3. They have described the collection, labeling and validation process

---

### Decision · Program_Chairs · 2021-07-27

**Decision:**

Reject

**Comment:**

The paper presents a dataset of quranic recitation audio paired with the text. The reviewers are concerned with the quality of the proposed benchmark, given the limitations mentioned in the paper. Most reviewers agree that the work could be improved with more description of the data collection process and more detailed analysis of crowd-sourced verification and automatic labeling. Also, they note that including more experiments with the potential use cases for this benchmark would make the paper more impactful. Given that the authors have not attempted to address the significant concerns raised by the reviewers, I recommend the rejection of this paper.